# Improving Odometric Model Performance Based on LSTM Networks

**DOI:** 10.3390/s23020961

**Published:** 2023-01-14

**Authors:** Bibiana Fariña, Daniel Acosta, Jonay Toledo, Leopoldo Acosta

**Affiliations:** Computer Science and System Department, Universidad de La Laguna, 38200 San Cristobal de La Laguna, Spain

**Keywords:** mobile robot, self-localization, odometry, sensor fusion, long short-term memory

## Abstract

This paper presents a localization system for an autonomous wheelchair that includes several sensors, such as odometers, LIDARs, and an IMU. It focuses on improving the odometric localization accuracy using an LSTM neural network. Improved odometry will improve the result of the localization algorithm, obtaining a more accurate pose. The localization system is composed by a neural network designed to estimate the current pose using the odometric encoder information as input. The training is carried out by analyzing multiple random paths and defining the velodyne sensor data as training ground truth. During wheelchair navigation, the localization system retrains the network in real time to adjust any change or systematic error that occurs with respect to the initial conditions. Furthermore, another network manages to avoid certain random errors by using the relationship between the power consumed by the motors and the actual wheel speeds. The experimental results show several examples that demonstrate the ability to self-correct against variations over time, and to detect non-systematic errors in different situations using this relation. The final robot localization is improved with the designed odometric model compared to the classic robot localization based on sensor fusion using a static covariance.

## 1. Introduction

Auto-localization is an essential component of autonomous robots. It enables the robot to navigate safely and accurately by providing real-time information about its pose and surroundings. To achieve this, auto-localization systems typically use fusion algorithms to combine data from multiple sensors, such as cameras, LIDARs, and GPS, providing a robust and accurate localization module even in dynamic environments. The fact of receiving several data based on different operating principles, allows the system to become resilient to sensor failure, ensuring that the robot can continue to navigate effectively.

There are many fusion techniques used in auto-localization, including the Kalman filter. This last one requires each sensor to provide a noise covariance, which describes its measurement accuracy and determines its influence on the final localization. For non-linear systems, there are variants of the Kalman filter, such as the Extended Kalman Filter (EKF) and the Unscented Kalman Filter (UKF). The EKF linearizes the system using the Taylor series, but it can be less effective in highly non-linear systems [1]. The UKF, on the other hand, uses the unscented transform to fit the probability density distribution of non-linear equations, which allows it to avoid the loss of higher-order terms that can occur with linearization [2]. This makes the UKF a simple, fast, and precise option for non-linear systems [3,4,5].

In order to achieve accurate state estimation for auto-localization systems, it is not enough to simply use the right fusion techniques. The reliability of the sensor data is also crucial. Sensors can be easily affected by noise, and overcoming this uncertainty is essential for improving the efficiency of the auto-localization.

One of the key sensors used in wheeled mobile robots is the odometric sensor, which is a precise, cheap, and easy-to-process sensor. However, the pose calculated by this sensor is incremental, which can lead to the accumulation of small deviations over time. This can result in significant errors in the pose estimation. Additionally, the odometric sensor is sensitive to random errors, such as drifting, which can make it difficult to track and potentially cause inaccurate poses.

Considering these limitations and how artificial intelligence is one of the fastest-growing technologies in mobile robotics, this paper presents the application of neural networks to odometry, improving the data accuracy and also reducing the sensor errors. In this way, the encoder data are processed by a neural network to include effects that a static model does not represent.

This method is implemented in an autonomous wheelchair that has been designed to transport people with severe disabilities [6]. The wheelchair is controlled by the user indicating the destination, while the navigation module will automatically execute the optimal path, using the sensors for the reconstruction of the environment and the obstacles detection [7]. In this way, to achieve safe autonomous navigation, the localization is an essential part of the chair movement, since its failure would affect all other subsystems.

The wheelchair has the following set of sensors:Odometry: it estimates the wheel robot movement by optical encoders coupled to the motor. In the prototype, the resolution of the encoder is 8800 pulses per revolution with a resolution of 0.04 degrees. The pose is calculated by the wheel movement knowing the radii and distance between them.Light Detection and Ranging (LIDAR): they are two Sick TiM 551 sensors, with a maximum distance of 10 m, an angular resolution of 1 degree and a viewing angle of 270 degrees. The relative movement of the robot can be calculated using two consecutive laser scans. The measurement accuracy is estimated by the ICP (Iterative Closest Point) algorithm [8,9]. It will return a low covariance value if two consecutive sweeps have many singular points that ensure their precise matching. However, if they have few singular points, the covariance will be high.Inertial Measurement Unit (IMU): an MPU9250 sensor which consists of an accelerometer, a magnetometer and a gyroscope. Our tests indicate that the magnetometer an the accelerometer are too imprecise to consider its measurements so only the gyroscope will be used. Likewise, the sensor covariance is estimated by setting a static value.

The robot state ${\tilde{x}}_{k}\left(V,W\right)$ is estimated by a UKF. In this case, the algorithm estimates the state at the next instant (k) by using the uniform rectilinear motion model and considering the sensor data $z_{k}\left(V,W\right)$ and their covariances ${\left(COV_{k}^{v/w}\right)}_{z}$ as measurements. The output of the filter not only includes the estimated robot state but also its covariance ${\left(COV_{k}^{v/w}\right)}_{filter}$, which measures the precision of the filter at that moment. Moreover, the pose (X,Y,θ) is calculated by integrating the output state of the filter. Figure 1 shows the wheelchair and a schema of the implemented localization where our technique is applied.

Accordingly, our paper focuses on the following key points: the robot self-localization problem applied to autonomous wheelchair and the LSTM (long short-term memory) neural network technique to improve the odometric sensor. The paper is structured as follows. Section 2 includes a review of the studies that have been completed prior to this work. Section 3 describes the wheelchair kinematics model. Likewise, Section 4 explains the methodology carried out to improve the accuracy of the odometric sensor. The experiments using the designed model are shown in Section 5, and the conclusions are summarized in Section 6.

## 2. Previous Work

The study of assistive robots is gaining attention for the possible social, economic, and scientific applications. There are investigations that collect the design of different types of autonomous wheelchair. Ref. [10] designs an autonomous wheelchair capable of segmenting passable areas and anomalies on the roads through deep learning. Refs. [11,12] show the characteristics of two wheelchairs for the transport of people in predefined hospital. Ref. [13] describes an adaptive neural control system for governing the movements of a robotic wheelchair. Likewise, [14,15] use an ROS (Robotic Operative System) in the wheelchair for the architecture of their modules.

Regarding novel techniques in the current literature on self-localization, there have been significant advances in the application of artificial intelligence (AI) to mobile robotics. This has led to the development of more intelligent and autonomous robots that can navigate complex environments and complete tasks more efficiently and accurately. Ref. [16] describes a survey of the different matching learning techniques for motion planing and control for mobile robots. In [17,18], a track fusion algorithm based on the LSTM method are proposed achieving better results in the fusion effect. Likewise, Refs. [19,20] present a deep reinforcement learning approach for the motion planning of autonomous robots in dynamic surroundings.

Machine learning is also applied in the odometry pose estimation to improve the results of using the traditional method by geometric equations. In [21], a feedforward neural network model is used in order to learn the odometric model from data. However, these sensors contain errors that need to be corrected in order to obtain a reliable pose. Real-time calibration of the structure’s parameters is important to adapt to any changes in the model over time. There are various techniques in the literature that attempt to adjust these parameters. Refs. [22,23] solves the car odometric systematic errors by using the final pose difference in a predefined path. Other works try to correct them in real time by the design of an AKF (Augmented Kalman Filter) [24] or by estimating the deviation of the radius nominal value by using a marginalized particle filter (MPF) [25].

With respect to non-systematic errors, such as slip, there are some methods that attempt to reduce them. Ref. [7] uses a Doppler speed sensor in order to measurement the odometric noise depending on the data quality and obtain a dynamic odometric covariance. Likewise, Refs. [26,27] try to correct them by reducing their influence in the output filter. They design a real-time outlier detection in the observations, applying a saturation function to the residual. Moreover, Refs. [28,29] try to avoid them by using the dynamic structure model. This last one proposes an slip ratio estimator based on the motion equations, the input torque and encoders data.

Taking into account the AI advances in the field of self-localization and the previously mentioned limitations of the odometric sensor, this paper presents a new approach based on LSTM networks for predicting odometry poses. The proposed system is able to self-calibrate in real-time through network retraining and another LSTM network is used to learn and compensate for non-systematic errors in the final localization.

## 3. Robot with Differential Kinematics

The wheelchair has differential kinematics. It consist of two drive wheels on a common axis. The robot can move and turn thanks to the independent wheel driving forwards or backwards. Figure 2 shows the scheme of a differential mobile robot with radii $R_{r/l}$ and a distance *D* between them. The robot movement between two time instants would correspond to a circle trajectory with radius ρ around the point ICC (Instantaneous Centre of Curvature). From the wheel speeds $V_{r/l}$, it is possible to estimate the linear (*V*) and angular (*W*) robot velocities, as well as its pose (X,Y) and its orientation (θ).

Figure 2 shows the robot wheel dynamics, where *T* corresponds to the motor torque that generates a linear $\left(V_{r/l}\right)$ and angular $\left(W_{r/l}\right)$ velocities. A friction force $\left(F_{r}\right)$ opposes to the movement, it can have a significant impact on the behavior and movement of the robot. These forces depend on the environment in which the robot operates, and small variations in their values can affect the robot’s movement. For example, certain types of robots, such as wheeled mobile robots, are very sensitive to small changes in the speed of each wheel. Even minor errors in the relative speeds between the wheels can affect the robot’s path. These robots are also sensitive to small variations in the ground plane, and they may need additional wheels (swivel casters) for support.

### 3.1. Wheel-Encoder-Based Traditional Odometry

The wheelchair’s pose at each instant can be obtained from the wheel speeds following the scheme of the Figure 2 by using odometric equations. In this case, we can estimate the respective wheel speeds by using the counts provided from the encoder sensor as follow,
(1)$$\begin{array}{ccc} V_{ok}^{l}=\frac{2\pi R_{l}▵ct_{k}^{l}}{encRes▵T}; & \; & V_{ok}^{r}=\frac{2\pi R_{r}count_{k}^{r}}{encRes▵T}\; \end{array}$$
where $V_{ok}^{l/r}$ corresponds to the left and right wheel velocities. $▵ct_{k}^{l/r}$ are the number of each encoder ticks that the electronics receive in time period (▵T), and encRes is the encoder resolution in one turn. The wheels parameters are the corresponding radii $R_{r}$ and $R_{l}$ and the distance between the wheels *D*.

Considering the circle trajectory with radius ρ around the ICC point, we can estimate the linear $\left(V_{k}\right)$ and angular ($W_{k}$) robot velocities (Equation (Equation 2)). This calculation is obtained by geometric equations knowing that the linear velocity of a circumference multiplied by the radius is equal to the angular velocity.
(2)$$\begin{array}{cc} \; & \frac{V_{ok}^{r}}{r}=W_{k};\;\frac{V_{ok}^{l}}{\rho +D}=W_{k};\;\frac{V_{k}}{\rho +D/2}=W_{k}\;\\ \; & W_{k}=\frac{\left(V_{ok}^{l}-V_{ok}^{r}\right)}{D};\;V_{k}=\frac{\left(V_{ok}^{l}+V_{ok}^{r}\right)}{2} \end{array}$$
Once the current robot velocities ${\left(V,W\right)}_{k}$ has been estimated, it is necessary to integrate, considering the time interval ▵T, to obtain the estimated robot pose ${\left(X,Y,\theta \right)}_{k+1}$ at each moment,
(3)$$\begin{array}{ccc} \theta_{k+1} & = & \theta_{k}+W_{k}▵T\\ X_{k+1} & = & X_{k}+V_{k}cos\left(\theta_{k}\right)▵T\\ Y_{k+1} & = & Y_{k}+V_{k}sin\left(\theta_{k}\right)▵T \end{array}$$

### 3.2. Wheel-Motion Pattern Effect

The behavior of wheeled mobile robots depends on the environment characteristics. The contact type between the wheel and the ground and its friction force value significantly influence the robot’s movement.

In ideal conditions, regardless of the terrain type, the robot speed depends on the motor torque and its power. An increase in the motor power will lead to an increase in angular wheel speed that will be translated into linear velocity. The opposite case is given to reduce the robot speed. However, there are situations where, depending on the current robot speed and the adhesion degree with the terrain, the robot can slip or skid, producing two situations. The first is due to the motor’s dead zone that does not produce enough torque to move the wheel, causing the wheel to stall instead of turning with the robot in motion. The second occurs when the wheel rotates freely, slipping and without any robot movement.

From this assumption, we anticipate that there is a logical correlation between the angular wheel speed and the motor power consumed. This relation disappears when non-systematic errors occur due to the type contact with the ground.

Figure 3 shows graphically an example of the relationship between the motor power and its angular velocity, given by the encoder sensor counts. In Figure 3a, there is no significant slip, following wheel speed and motor power, a similar pattern, increasing and decreasing in a correlated way. However, in Figure 3b there is a slip, highlighted by the green lines, where, although the power increases, the real angular velocity decreases, contrarily to the expected effect. Using this property, the motor power can be used to detect non-systematic errors in the odometric system. There is not a clear mathematical relation between motor power and speed in order to detect non systematic errors, so a Neural Network is used to learn this relation from data.

## 4. Odometry System Proposed

In this localization system, the data accuracy provided by the odometric sensor is improved using neural networks that learn from the robot behavior. It avoids using manually estimated parameters, improving and correcting some errors, which in the mathematical model are not appreciable and, therefore, difficult to adjust.

The proposed model consists of two neural networks that have, as input, the incremental encoder counts $\left(▵ct_{k}^{r/l}\right)$ of the right and left wheels, and provides, as output, the robot state, that are the linear $\left(V_{k}\right)$ and angular $\left(W_{k}\right)$ speeds for the respective network. We decided to use two neural networks for each output instead of a single one in order to allow each network to specialize in predicting a specific output and converge more quickly and easily. This is because in the following sections we will have to retrain them online and we need the network to quickly adapt and converge to any changes that may occur in the model. Using a single network with two outputs makes retraining much more difficult and it does not adequately adjust to our desired requirements. This has also been tested experimentally, and we found that the architecture proposed resulted in more precise execution.

The training has been carried out offline by creating a dataset from the information of multiples paths with different characteristics. During the training, the speed data provided by a velodyne HDL 32 sensor at each moment have been collected as the desired output of the robot state. It has been installed in the upper part of the chair and consists of 32 LIDARs with a precision of 2 cm that enables the calculation of the state with high reliability using the LOAM slam algorithm [30,31].

We decided to use recurrent networks, specifically, the LSTM-type [32] was chosen over other models, such as MLP, because of its ability to learn and remember patterns over time. This is particularly useful as it can take into account long-term patterns in the encoder data sequence. Moreover, during the neural network design, we compared the use of other models using the wheelchair dataset and it demonstrates a better effectiveness during training and validation. For the LSTM network, it has been considered a time window of 20 samples that are provided by the sensor at a frequency of 10 Hz. Figure 4 shows the structure carried out to obtain the linear $\left(V_{k}\right)$ and angular $\left(W_{k}\right)$ velocity at each time *k*. In order to train the designed model, pre-processing data are necessary. First, the raw data were resized into a tensor whose dimension was (samples, time step, features), which in our case was (10,115,20,3). The inputs at one moment correspond to the encoder count increments and the speed at the previous time ${\left(▵ct^{r},▵ct^{l},V/W\right)}_{k-1}$. The outputs are the lineal $V_{k}$ and angular $W_{k}$ speeds of the robot. Moreover, the input must be normalized to the same scale by eliminating the mean and scaling to a standard deviation equal to 1. So, the samples of an input type are pre-processed using:(4)$$\begin{array}{c} z=\frac{s-\mu }{\sigma }\;\;with\;\;\mu =\frac{1}{N}\sum_{i=1}^{N}s_{i}\;\;\sigma =\sqrt{\frac{1}{N}\sum_{i=1}^{N}{\left(s_{i}-\mu \right)}^{2}} \end{array}$$
where *z* is the new normalized value and *s* is the raw sample. μ is the mean and σ is the standard deviation of the input, and *N* correspond to the samples dimension.

Regarding the implemented LSTM layer, it analyzes an input data stream and produces a prediction. In this structure, data are introduced sequentially in an LSTM unit (cell) that can be divided into different parts. In each cell, $x_{t}$ corresponds to the input at the current time, $h_{t-1}$ is the previous cell output and $C_{t-1}$ is the cell state. First, the forget gate controls how much information about $C_{t-1}$ is relevant from the inputs $h_{t-1}$ and $x_{t}$, by using a sigmoid layer (σ),
(5)$$\begin{array}{c} f_{t}=\sigma \left(\omega_{f}\left[h_{t-1},x_{t}\right]+b_{f}\right) \end{array}$$
where $\omega_{f}$ represents the weight matrix and $b_{f}$ the bias. $f_{t}$ value in each dimension is in the range (0, 1). The information will be forgotten when $f_{t}$ is close to 0, and the information will be retained when is close to 1.

Subsequently, the gateway is responsible for processing the current input and updating the relevant information. It has two parts, the first $\left(i_{t}\right)$ controls how much of the input is stored using a sigmoid layer and the second (${\tilde{C}}_{t}$) generates a new current state candidate by using the tanh function,
(6)$$\begin{array}{c} i_{t}=\sigma \left(\omega_{i}\left[h_{t-1},x_{t}\right]+b_{i}\right)\;\;{\tilde{C}}_{t}=tanh\left(\omega_{c}\left[h_{t-1},x_{t}\right]+b_{c}\right) \end{array}$$
where $\omega_{i}$ and $\omega_{c}$ represent weight matrix and $b_{i}$ and $b_{c}$ represent the bias. From this information, the cell state can be updated from the previous one $\left(C_{t-1}\right)$ and from the new generated candidate $\left({\tilde{C}}_{t}\right)$,
(7)$$\begin{array}{c} {\tilde{C}}_{t}=f_{t}*C_{t-1}+i_{t}*{\tilde{C}}_{t} \end{array}$$
Finally, the output gate generates the cell output $\left(h_{t}\right)$ from the selection of the relevant information in the current cell state,
(8)$$\begin{array}{c} h_{t}=\sigma \left(\omega_{o}\left[h_{t-1},x_{t}\right]+b_{o}\right)*tanh\left(C_{t}\right) \end{array}$$
where $\omega_{o}$ is the weight matrix and $b_{o}$ its bias.

Therefore, considering the Figure 4, the data are processed and normalized, then divided into a 70% of training and a 30% as test data. In this case, the LSTM is 1 layer with 5 neurons in each cell, chosen ad hoc, providing the best training and validation results for different tests. This state is passed to a full connected output layer with 1 neuron for a single output and linear activation. Regarding the training, the ADAM optimizer algorithm and the MAE (mean absolute error) metric has been used with a learning rate of 0.001, a batch size of 10, 150, and 200 epochs depending on the model. Figure 5 shows the loss in both training and validation, using the MAE errors obtained with respect to the epochs for each model, V and W.

### 4.1. Odometry Error Covariance Modeling

The odometric system can be improved using an LSTM network to detect non-systematic errors that occur in unpredictable environments. For example, local bumps, holes, rocks, and non-level terrain can cause non-systematic errors. Using gyro sensors to detect these errors has shown good performance, as demonstrated in previous research [33,34]. However, there are other cases that are more difficult to detect, such as wheel slippage on level terrain. In these cases, a non-systematic detection method is necessary.

As explained in the first section, the fusion algorithm estimates the robot localization from different sensors, where each measurement is defined with a covariance that characterizes its confidence. In the case of odometry, it takes the speed covariances ($COV_{k}^{V},COV_{k}^{W}{)}_{odom}$. Then, the filter prioritizes the measurements with a small covariance and avoids those with greater values. So, it is important to dynamically adjust the covariance of the odometric sensor to assign a high number in the presence of non-systematic errors and, thus, reduce its influence on the final localization result, compared to other sources. This variable covariance is designed based on the assumptions in Section 3.2, where we demonstrate how the wheel speeds depend on the motor torque and its power, and how the adhesion degree with the terrain causes the wheel to slip, affecting this dependence. Therefore, we need a system that is able to learn the relationship between the power of each wheel and its corresponding angular velocity in order to detect when this relationship is not fulfilled and is in an error state. In this case, the system will act accordingly by assigning a very large static value to all elements in the covariance matrix, considering that they are stochastically independent random variables.

Figure 6 shows the method implemented on the wheelchair for this purpose. It consists of an LSTM neural network that takes as input the powers of each wheel $\left(P^{r},P^{l}\right)$, the encoder increment counts $\left(▵ct^{r},▵ct^{l}\right)$, and the sensor covariance in the previous measurements ($COV_{k-n}^{V}$), $\left(COV_{k-n}^{W}\right)$. As output, it is able to estimate the next step covariance ($COV_{k}^{V}$), $\left(COV_{k}^{W}\right)$. The data have been preprocessed using the same method discussed earlier, organized into time windows with memory of 20 samples, and normalized to have a standard deviation of one and a mean equal to zero. In our case, several samples have been collected with different slip types and whose input dimension is (12442,20,5), which have been divided into 70% for training and 30% for validation. The training has been carried out offline by using the velodyne sensor data. The network structure is a LSTM layer consisting of 10 neurons and a full connected output layer with 1 neuron for a single output and ReLU activation. We considered the ADAM optimizer algorithm and the MAE metric with a learning rate of 0.001, 120 epochs, and a batch size of 10. A de-normalization process is required in the final phase to obtain the estimated covariance data.

Figure 6 also shows the variance of the angular odometry velocity estimated by our designed network with respect to time, where a slippage happens (non-systematic error) in the period 400–405 s. As we can see, the covariance remains at a low value when the sensor operates correctly and increases considerably when it is in an error state. This information allows the UKF filter to weight the input data in real time. If the LSTM Network detects a failure condition, the covariance will increase, so the data coming from odometry will be considered by the filter with a small weight. The odometry data together with this variable covariance will be used in a subsequent filter to be fused with other sensors and its influence on the final localization result will depend on that estimated covariance.

### 4.2. Real Time Auto-Calibration

Real-time calibration of the model is crucial when designing an odometry system. It must be able to detect and correct any systematic error that may arise in relation to the offline-trained model. For example, a sudden change in the model may occur due to a shift in the weight of passenger, which can significantly impact the model accuracy. Over time, other situations, such as a change in tire pressure, can also affect the model performance, causing errors in the estimated localization. In order to ensure high accuracy, the system should be able to adapt to these changes in real-time by fine-tuning its model.

The scheme used for this calibration is shown in Figure 7. In this case, the neural network models for both angular and linear velocity estimation are retrained in real-time using input data from encoders $\left(▵ct^{r},▵ct^{l}\right)$ and the output from the UKF filter speeds ${\left(V,W\right)}_{k}$. It is an iterative process where the data are stored in real-time, and when 300 samples are collected, the network weights are adjusted accordingly. These samples are only stored whenever the covariance of the filter $\left(COV_{Filter}\right)$ and the variable covariance of the odometry $\left(COV_{Odom}\right)$ are small, which ensures that the data used for re-training are reliable and accurate. In this way, the model is changed only when the robot is well localized, avoiding introducing errors in the retraining process.

Figure 7 also shows the error in angular velocity over time with respect to the ground truth (velodyne) when there is a change in the model, causing the error to increase. These graphs compare the behavior of the odometric system with and without re-training. Without considering the calibration, the model does not detect any change and the error grows and remains constant. In case of the real-time calibration, the system is able to detect these changes and adjust the model by retraining, reducing the relative error with respect to the velodyne sensor.

## 5. Results

The validation of the implemented wheelchair localization system was demonstrated through the study and analysis of several paths with different characteristics. These experiments were carried out in the corridor of the robotics laboratory at the University of La Laguna. In addition, to evaluate the proposed odometric model, different parameters and speeds were used on the paths to demonstrate the functionality of its variable covariance and the online calibration, as well as to ensure that a wide range of circumstances were covered. The effectiveness of the calibration was tested by varying the passengers and altering the diameters of the wheels through inflation. Furthermore, in the experiments to detect errors from the environment, the wheelchair speed has been increased in various sections and we have used other wheels with greater wear to generate slipping. We have also created these situations by applying pressure on one side, preventing the rotation of a wheel while allowing the other to turn freely.

On the one hand, Figure 8a,b shows two different example paths where the estimated pose of the odometry using LSTM networks (green line) and the pose obtained using traditional odometry (blue line) have been compared with the reference data from the velodyne (black line). As can be observed, the use of our model significantly improves the estimated pose accuracy, providing results that are closer to the ground truth.

This fact is also demonstrated numerically in Table 1. In this case, multitude of trajectories have been executed and the odometry error with respect to the velodyne has been collected for both cases, using our LSTM network and using traditional model. We have analyzed the errors at each time in the velocities (V,W) and in the robot pose (X,Y,θ). The table shows the mean squared error (RMSE), the mean absolute error (MAE), and the R squared, which are defined by the following formulae: (9)$$RMSE=\sqrt{\frac{1}{n}\sum_{i=1}^{n}{\left({\hat{y}}_{i}-y_{i}\right)}^{2}}\;MAE=\frac{1}{n}\sum_{i=1}^{n}\left|{\hat{y}}_{i}-y_{i}\right|\;R^{2}=1-\frac{\sum_{i=1}^{n}{\left({\hat{y}}_{i}-y_{i}\right)}^{2}}{\sum_{i=1}^{n}{\left(y_{i}-\bar{y}\right)}^{2}}$$
where ${\hat{y}}_{i}$ is the predicted value, $y_{i}$ is the true value, *n* is the number of samples, and $\bar{y}$ is the mean value of the true values. Based on the obtained values, the errors of our system are smaller than the traditional model. Therefore, despite the small difference between the errors, it demonstrates the effectiveness of our model in improving the odometry localization of the wheelchair.

On the other hand, the proposed real-time calibration model has been studied in several paths and with different changes in the system that have occurred both initially and during it. Figure 9 shows two examples where the characteristics of one wheel have been intentionally modified from the initial model to represent the odometry behavior when a systematic error arises. It causes the pose to deviate, generating an error in the velocities that, when integrated, accumulates over time in the pose. We compared the behavior of three different odometry models: the neural network-based odometry with real-time retraining (green line), odometry with neural networks but without retraining (yellow line), and traditional odometry (blue line). As reference, we consider the velodyne(black line).

This figure shows how, thanks to retraining, the model is adjusted to new changes and corrects the pose increments of the trajectory, approaching the velodyne. Table 2 collects the errors obtained from several experiments comparing the proposed odometry model with and without retraining. Based on these results, the neural network-based odometric model and traditional odometry are unable to perceive these changes, increasing the error indefinitely. If we consider the calibration, the error in the velocities is reduced. However, it does not directly correct the error pose, but rather gradually improves the pose increments at each time interval. Moreover, we are interested in correcting the velocities, which will be the input measurement data for the filter in estimating the final pose of the robot.

Regarding the process implemented to reduce the influence of non-systematic errors, the output of the UKF filter has been studied in various experiments with different slip types. Figure 10 shows a path example where a considerable non-systematic error occurs. The green line represents the pose estimated by the odometry, the blue one the LIDAR, the red is the UKF localization, and the black line defines the ground truth. This error type causes the odometry pose to deviate significantly, and since it is a random error, it is impossible to correct its pose. The proposed solution is to modify the odometry covariance. During the corridor, the LIDAR sensor has a high covariance due to the low number of possible matching points and its estimated pose could be erroneous, however the odometry provides accurate results. In turn, due to the slip, the error in the odometric sensor grows and the LIDAR is more accurate.

Figure 10a shows the UKF-filter output when the traditional odometry is used with a static covariance manually adjusted. The result is worse since it either relies too much on odometry, underestimating the LIDAR or vice versa. Figure 10b shows the localization result of the UKF filter with the proposed LSTM neural network with non-systematic error detection and dynamic covariance estimation. During the slip, the power of the motors and wheel speed is not correlated, the error classifier detects a non-systematic error during the turn, and increases the covariance, reducing the odometry influence on the final localization during that error state. The final pose of the filter is close to ground truth, and clearly better than without the detector.

In addition to the graphical study, the errors of the UKF filter, with respect to the velodyne, have been collected in different experiments for both cases, using traditional odometry with static covariance and the proposed odometry with variable covariance. Table 3 shows the metric values used for its comparison that are MAE, RMSE, R squared, and we have also considered the normalized estimation error squared (NEES) using the filter results and the estimated odometry poses against the ground truth. This metric takes into account the state covariance and is defined as:(10)$$\begin{array}{c} NEES=\frac{1}{n}\sum_{i=1}^{n}{\left(x_{i}-{\hat{x}}_{i}\right)}^{T}P_{i}^{-1}\left(x_{i}-{\hat{x}}_{i}\right) \end{array}$$
where *n* is the number of samples, $x_{i}$ is the true state, ${\hat{x}}_{i}$ is the estimated state, and $P_{i}$ is the covariance matrix for the *i*th sample.

Therefore, the proposed odometry model with a variable covariance has been shown to improve the accuracy of the UKF filter in the presence of non-systematic errors. This has been demonstrated through graphical analysis and by comparing the errors of the UKF filter using the proposed odometry model and the traditional odometry model with static covariance. The NEES metric has also been used to evaluate the accuracy of the filter, with a lower NEES value indicating better performance.

## 6. Conclusions

This work presents a new odometry system implemented in an autonomous wheelchair. It consists of LSTM neural networks that are able to estimate the robot speed using the data from the encoder sensor. The real-time retraining allows the system to self-calibrate and adapt to changes in the defined model, further improving its performance. Likewise, it is able to reduce the influence of some non-systematic errors by training LSTM networks that learn the relationship between the power of the wheels and their angular velocity to design a variable covariance.

In conclusion, this system significantly improves the accuracy of the wheelchair estimated pose by the UKF filter, compared to the use of classic odometry methods. This is demonstrated through graphical analysis and numerical comparisons of the errors between the different methods. The results show how our system errors are smaller and closer to the velodyne data, providing more accurate final robot localization, particularly in presence of systematic and non-systematic errors. Overall, the proposed wheelchair localization provides a more robust solution for state estimation in challenging environments.

## Figures and Tables

**Figure 1 sensors-23-00961-f001:**
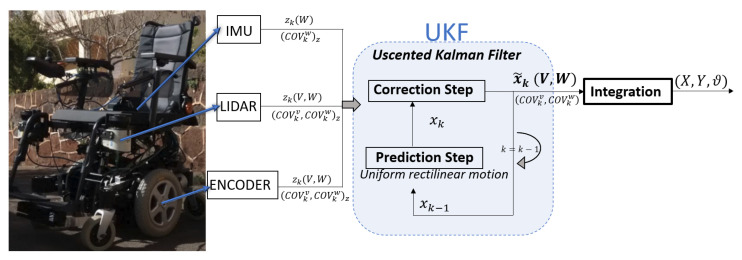
The localization system proposed for the wheelchair.

**Figure 2 sensors-23-00961-f002:**
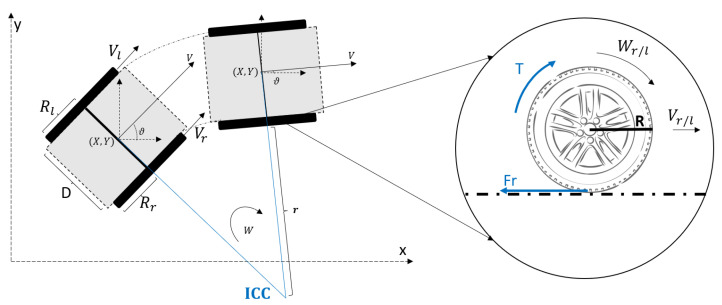
Differential kinematics schema for a mobile robot.

**Figure 3 sensors-23-00961-f003:**
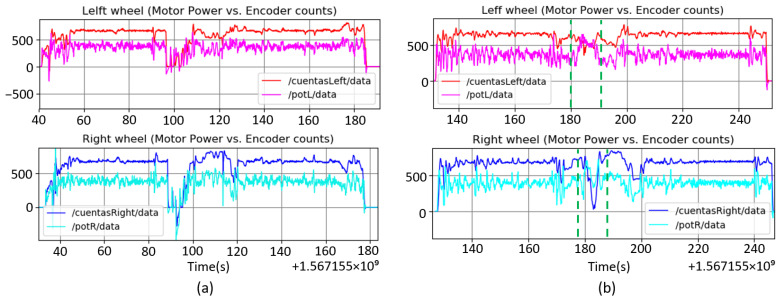
Motor power vs. encoder counts for both wheels.(**a**) Without slipping and (**b**) with slipping.

**Figure 4 sensors-23-00961-f004:**
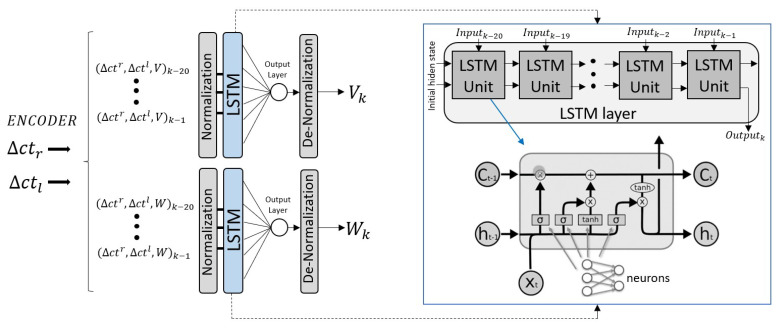
Schema of the odometric model proposed using LSTM networks.

**Figure 5 sensors-23-00961-f005:**
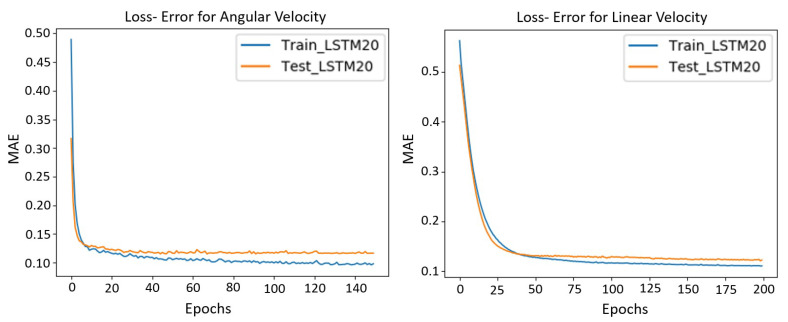
Loss function per epochs using the MAE for the angular and linear velocities.

**Figure 6 sensors-23-00961-f006:**
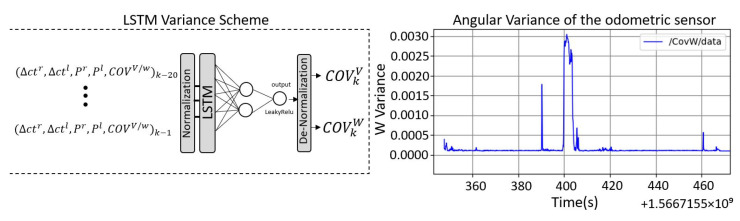
Scheme implemented to avoid non-systematic odometry errors, and the evolution of the angular velocity variance when a slippage happens in an example path.

**Figure 7 sensors-23-00961-f007:**
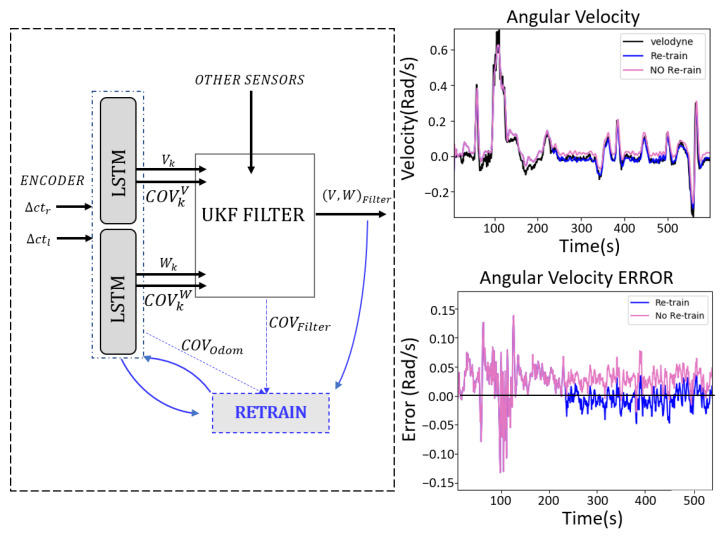
Retraining scheme for odometric model calibration. The error using the model with real-time retraining is smaller than using the model without it.

**Figure 8 sensors-23-00961-f008:**
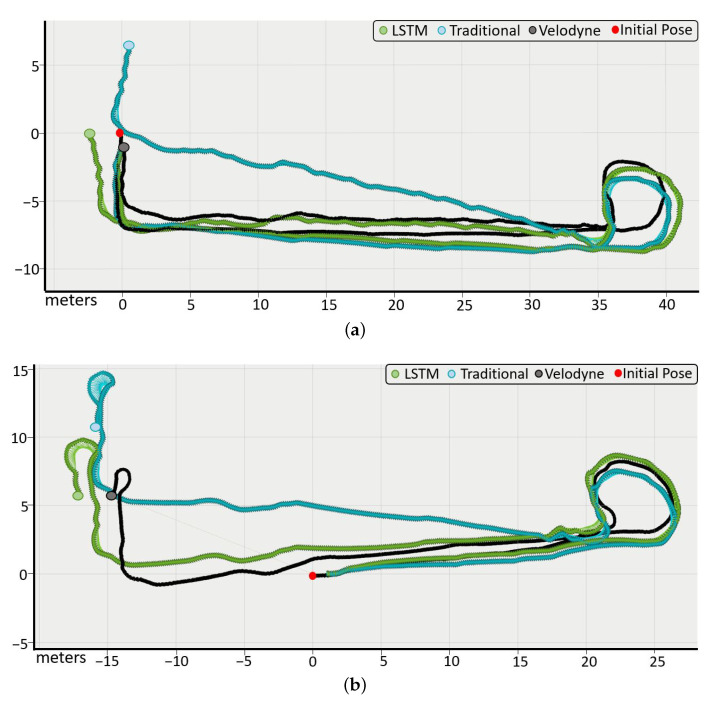
Paths to validate the odometry model proposed. Odometry with LSTM networks (green), traditional odometry (blue), and ground truth (black).

**Figure 9 sensors-23-00961-f009:**
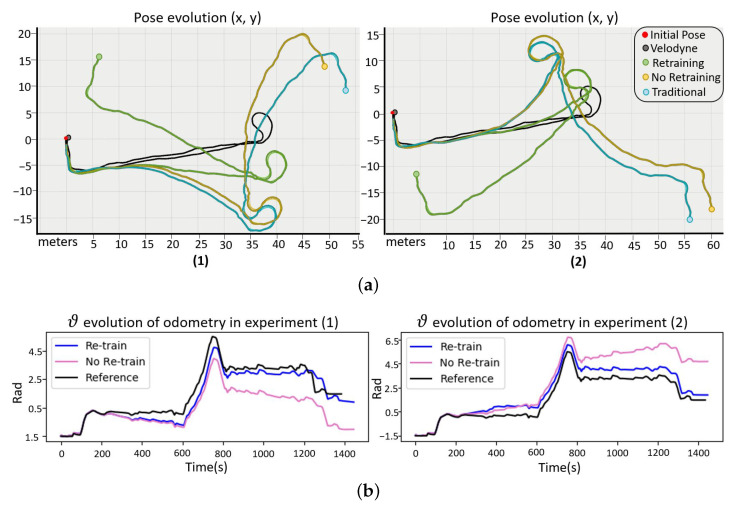
Example paths to validate the calibration model by retraining in real-time. (**a**) Shows the position “X, Y” and (**b**) shows the orientation “θ”.

**Figure 10 sensors-23-00961-f010:**
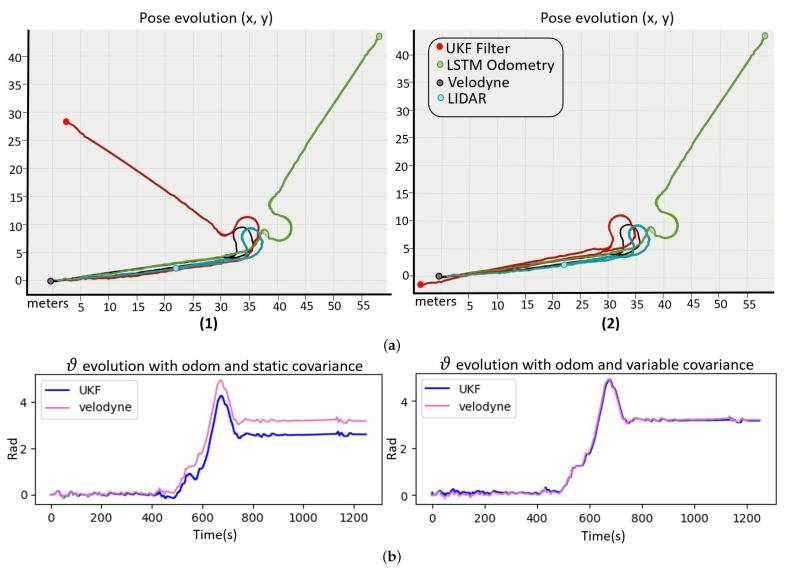
Localization system for a path where the red line is the filter output, the green is the odometry, the blue the LIDAR, and the black line for the ground truth. (**a**) Shows the position “X, Y” and (**b**) shows the orientation “θ”. (**1**) Odometry with static covariance and (**2**) odometry with variable covariance.

**Table 1 sensors-23-00961-t001:** Error values obtained by comparing the LSTM odometry model and the traditional odometry with respect to the velodyne data.

		V (m/s)	W (rad/s)	Pose	
				X,Y (m)	θ (rad)
	MAE	0.04646	0.08103	1.56692	0.07131
LSTM odometry	RMSE	0.10282	0.1791	2.076499	0.0864
	$R_{score}$	0.799122	0.84129	0.9569	-
	Accuracy %	79.91%	84.13%	95.69%	-
	Global Model Accuracy (%)			86.57%	
	MAE	0.062744	0.102723	1.83725	0.1720
Traditional odometry	RMSE	0.1237	0.200329	2.48464	0.2151
	$R_{score}$	0.6734	0.7953	0.9389	-
	Accuracy %	67.43%	79.53%	93.89%	-
	Global Model Accuracy (%)			80.2%	

**Table 2 sensors-23-00961-t002:** Error values obtained by comparing the LSTM-odometry model with and without retraining considering the velodyne data as reference.

		V (m/s)	W (rad/s)	Pose	
				X,Y (m)	θ (rad)
	MAE	0.0384577	0.06484	4.27604	0.56772
LSTM Retraining	RMSE	0.069028	0.115653	8.34219	0.629522
	$R_{score}$	0.848956	0.71317	0.25404	-
	Accuracy %	84.9%	71.32%	25.4%	-
	Global Model Accuracy (% )			60.54%	
	MAE	0.05496	0.07919	5.156277	1.5677
LSTM No retraining	RMSE	0.081812	0.22197	8.90407	1.7094
	$R_{score}$	0.67216	0.57198	0.03743	-
	Accuracy %	67.22%	57.2%	3.74%	-
	Global Model Accuracy (%)			42.7%	

**Table 3 sensors-23-00961-t003:** Error values obtained by comparing the UKF filter output with respect to the velodyne data in two cases. Using the odometry model with variable covariance and with static covariance.

		V (m/s)	W (rad/s)	Pose	
				X,Y (m)	θ (rad)
	MAE	0.04125	0.03149	1.4021	0.0547
LSTM odometry	RMSE	0.03804	0.05464	2.8699	0.07037
	$R_{score}$	0.83329	0.80617	0.9279	-
	$NEES_{Odom}$	1087	1982	-	-
	$NEES_{Filter}$	1614	2963	-	-
	Accuracy %	83.33%	80.62%	92.8%	-
	Global Model Accuracy (%)			85.58%	
	MAE	0.05333	0.0452	2.0592	0.38356
Traditional odometry	RMSE	0.0661	0.071619	5.3323	0.4759
	$R_{score}$	0.7697	0.67634	0.678483	-
	$NEES_{Odom}$	6439	7650	-	-
	$NEES_{Filter}$	5986	4779	-	-
	Accuracy %	76.97%	67.63%	67.85%	-
	Global Model Accuracy (% )			70.82%	

## Data Availability

Not applicable.

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
