# Peer review of "Improving Odometric Model Performance Based on LSTM Networks"

_sensors, 2023, doi:10.3390/s23020961_

Round 1

Reviewer 1 Report

- Figure 4: why a neural network for each output. The two outputs are linked somehow I think that a single neural network with two outputs gives a better result.

- If we choose a single neural network that accepts as input the encoder parameters (and the two voltages of two motors or the difference between the two quantities) and as output the situation of the robot, by this way we take into account the asymmetry caused by the nature of the environment and whatever its type and at the same time we simplify the problem of generating the database (learning and validation). In addition, we don't need to perform the e-learning which cause a lot of problem in practice. If  architecture described above is not suitable, please explain why.

- The results given by Figures 9 and 10 are not well interpreted. There are significant errors between the final situations. 

- Figures 9 and 10 give no information on the orientation of the robot.

Reviewer 2 Report

Improving odometric model performance based on LSTM networks by Bibiana Fariña, Daniel Acosta, Jonay Toledo and Leopoldo Acosta.

Regarding the reviewed work, I have no major remarks or concerns. The paper is well organized and presented in clear detail. The advantages of the proposed new odometry system provide more accurate final robot localization. Compared to classical odometry methods, the UKF filter significantly improves the accuracy of the wheelchair's estimated pose.

Minor comments:

Why was the LSTM neural network used instead of another training model?

Page 9, line 312. It would be good to expand and better explain exactly how the experiments were carried out. How these "wide range of circumstances" were created.

Increase the font size of the ticks on both axes in all figures.
